# Overemphasis on publications may disadvantage historically excluded groups in STEM before and during COVID-19: A North American survey-based study

**Freya E. Rowland**[¤a]*[☯], **Kyra A. Prats**[¤b☯], **Yara A. Alshwairikh, Mary K. Burak, Ana Clara Fanton**[¤c], **Marlyse C. Duguid**

School of the Environment, Yale University, New Haven, Connecticut, United States of America

☯ These authors contributed equally to this work.
¤a Current address: U.S. Geological Survey, Columbia Environmental Research Center, Columbia, Missouri, United States of America
¤b Current address: Department of Botany and Plant Pathology, Purdue University, West Lafayette, IN, United States of America
¤c Current address: Institut National de Recherche pour l'Agriculture, l'Alimentation et l'Environnement (INRAE), Villenave d'Ornon, France
* frowland@usgs.gov

**Data Availability Statement:** The data that support the findings of this study are openly available in Dryad at https://doi.org/10.5061/dryad.fqz612jwn.

## Abstract

Publishing is a strong determinant of academic success and there is compelling evidence that identity may influence the academic writing experience and writing output. However, studies rarely quantitatively assess the effects of major life upheavals on trainee writing. The COVID-19 pandemic introduced unprecedented life disruptions that may have disproportionately impacted different demographics of trainees. We analyzed anonymous survey responses from 342 North American environmental biology graduate students and postdoctoral scholars (hereafter trainees) about scientific writing experiences to assess: (1) how identity interacts with scholarly publication totals and (2) how the COVID-19 pandemic influenced trainee perceptions of scholarly writing productivity and whether there were differences among identities. Interestingly, identity had a strong influence on publication totals, but it differed by career stage with graduate students and postdoctoral scholars often having opposite results. We found that trainees identifying as female and those with chronic health conditions or disabilities lag in publication output at some point during training. Additionally, although trainees felt they had more time during the pandemic to write, they reported less productivity and motivation. Trainees who identified as female; Black, Indigenous, or as a Person of Color [BIPOC]; and as first-generation college graduates were much more likely to indicate that the pandemic affected their writing. Disparities in the pandemic's impact on writing were most pronounced for BIPOC respondents; a striking 85% of BIPOC trainees reported that the pandemic affected their writing habits, and overwhelmingly felt unproductive and unmotivated to write. Our results suggest that the disproportionate impact of the pandemic on writing output may only heighten the negative effects commonly reported amongst historically excluded trainees. Based on our findings, we encourage the academy

All associated code is publicly archived on Zenodo for permanent access (https://doi.org/10.5281/zenodo.8256672).

**Funding:** FER was supported by the Yale Institute for Biospheric Studies (YIBS) Donnelley Postdoctoral Fellowship. They did not specifically fund this work, but did pay for FER's fellowship.

**Competing interests:** The authors have declared no competing interests exist.

to consider how an overemphasis on publication output during hiring may affect historically excluded groups in STEM—especially in a post-COVID-19 era.

## Introduction

Learning the nuances of academic writing is nonlinear and writing challenges can emerge at any point in one's career. Graduate students and postdoctoral scholars (i.e., trainees) are learning how to communicate their research through writing manuscripts for publication. For trainees seeking tenure-track positions, successful applicants are often those who have published the most [1,2]. The number of publications by recently hired trainees has doubled over time [3] and high publication totals are critical for securing academic employment [4]. This has created a feedback loop where increased competition among job applicants has led trainees to feel pressure to increase their own publication totals in order to secure academic research positions.

Despite broad awareness of the importance of academic publishing, there is evidence that writing output is not equal across identity groups. The range of affiliations that one holds, or identity (Box 1) can be associated with differential access to writing support and output. Students from historically excluded groups in STEM (Box 1) face many barriers to accessing higher education [5], such as difficulty obtaining support and guidance for applying to college —a process that involves significant writing. This differential access to writing support extends throughout undergraduate years, and likely persists in graduate school as well. Prior to the COVID-19 pandemic (hereafter the pandemic), research showed that historically excluded (e.g., Black, Indigenous, People of Color [BIPOC] (Box 1), women) graduate students in STEM fields were less likely to publish than white male graduate students [6]. Many graduate students—especially historically excluded groups (HEGs, Box 1) and those whose first language is not English—experience anxiety while writing [7,8]. Researchers with disabilities are funded at lower rates than those without a disability [9] and many report lacking support from their institutions [10].

### Box 1. Terminology

We have strived to be as inclusive as possible throughout the process of creating and distributing our survey, as well as through our choice of language. When discussing **identity** as a whole, we used the American Psychological Association [11] definition as "an individual's sense of self defined by (a) a set of physical, psychological, and interpersonal characteristics that is not wholly shared with any other person and (b) a range of affiliations (e.g., ethnicity) and social roles." Here we describe the identity terminology that we chose to use in our survey and writing. To remain consistent, we use these terms throughout our writing.

We asked survey respondents to self-identify their gender as "**female**," "**male**," "**nonbinary/third gender**," or they could select "**prefer to self-identify**." Thus, when referring to female and male genders we are not implying the biological sex of individuals, but rather employing the terms as adjectives of their gender identity.

We use **Black, Indigenous, and people of color (BIPOC)** to refer to groups that have faced and are facing racism. Where possible, we try to center all these groups rather than using the term non-white, which still centers white people. We recognize that BIPOC individuals come from diverse and unique backgrounds, and therefore do not all have the same experience. In the survey, individuals specified whether they identified as BIPOC or not.

The term individuals with a **chronic condition** describes individuals with a chronic health condition or disability. We understand that many people within the disability community prefer person-first language (e.g., person with a disability), while others prefer identity-first language (e.g., disabled person). We use the term individuals with a chronic condition to be inclusive of all types of disability and chronic health conditions.

Trainees with **English as a second language (ESL)** were those who did not have English as their first language. We use the term ESL to encompass trainees who speak English as a foreign language (EFL) and English as an additional language (EAL). We did not differentiate international trainees (i.e., those studying or working in countries that are not their native country) and in-country trainees with ESL, but we recognize that many respondents with ESL may also be international trainees.

**Historically excluded groups (HEGs)** refers to any group of people who have been excluded from full rights and privileges based on historical systems of oppression. This can include (but is not limited to) female individuals, transgender individuals, non-binary or third gender individuals, BIPOC individuals, and individuals with chronic health conditions. We chose not to use the term 'underrepresented minority' as the term 'historically excluded' better encompasses the power dynamics and systems of oppression that governed which groups were excluded. The term HEG is also commonly used in publications [12–14].

There is a gap in the literature regarding how trainees from other HEGs—such as first-generation college students and gender non-binary trainees—may be impacted by the pressure to publish. The studies that do exist record experiential challenges but do not quantitatively explore how identity may influence trainee publication outputs. The importance of increasing diversity within academia is recognized [15], yet faculty hiring is hierarchical and unequal. For example, women are often hired by less prestigious universities than male peers from the same doctoral institutions [16]. Women of color, whose experiences intersect at race and gender [17,18], are underrepresented in higher levels of academia [19]. Furthermore, some trainees from HEGs may spend significant time on service towards improving Justice, Equity, Diversity, and Inclusion (JEDI) initiatives within academia. Post-graduation, many women of color reported a strong desire to engage in activism, increase diversity, and improve conditions for other BIPOC women in STEM [18]. BIPOC women faculty have reported that engaging in activism—rather than focusing solely on research—led to forfeiting a full professor rank but felt advocating for HEGs was more important [18].

Increasing diversity in academia necessitates fair assessment of scholars. Within the current academic system, the main metric of success is tangible outputs such as publications and grants [20]. There have been multiple internal calls for change, particularly for being held accountable for how we achieve outputs (e.g., being a good leader, teacher, and mentor, without harming other community members; [15]) and prioritizing equity for our community members [15,21–23]. Academic communities must also understand how life experiences affect trainee publishing output, including how identity may intersect with major life upheavals. Although recent publications highlight how the pandemic affected faculty [24,25], there is little understanding of how it impacted trainee writing productivity. Here, we (1) conducted a survey to contextualize how the pandemic and identity may interact to disproportionately affect historically excluded trainees, and (2) discuss what equitable metrics of success look like in academia.

## Materials and methods

### Survey design and distribution

During March–April 2021, we used an anonymous Qualtrics survey to ask academic trainees (i.e., graduate students and postdoctoral scholars [hereafter postdocs]) currently working at American and Canadian Universities within environmental biology fields to self-identify their demographic information, current peer-reviewed publication records, and the effects of the pandemic on their feelings towards writing habits and productivity (full survey available in S1 File). We chose to restrict the sample frame to environmental biology trainees because STEM sub-fields can have large differences in advisor and department involvement and expectations for publishing [6].

We were approved by Yale University's Institutional Review Board as exempt human subjects research under 45CFR46.104 (2)(ii) and did not track any personal identifying information or geographic location (e.g., IP address) to allow for honest and open answers. Participants gave written informed consent at the start of the survey (full survey in S1 File). Survey completion was voluntary, and we advertised it via social media (Twitter, Reddit, Facebook, Instagram), targeted emails to colleagues, and posted the survey twice on the ECO-LOG-L listserv hosted by the Ecological Society of America. To increase the geographic diversity of the sample frame, we emailed 98 different department chairs or graduate coordinators from at least one major public R1 university in each U.S. state and Puerto Rico, U.S. Ivy League institutions, and five R1 Canadian Universities and asked them to distribute our survey among trainees. It is probable that trainees from non-R1 institutions filled out our survey after finding it on social media; however, we cannot determine the percentage of trainees from R1 or non-R1 universities. We did not require respondents to answer all questions; therefore, some respondents skipped questions or left fields blank. We eliminated these blank or skipped responses and categorized them as "NA", but as a result, the sample size differs between questions. We denote sample size either in text or in figure captions to account for this.

### Data analysis

We fit Bayesian multiple linear regressions in R version 4.0.2 (R Core Team 2020) using 'rstanarm' [26] to estimate how training years and identity (see Box 1) affected total publications using a Gaussian distribution. We estimated whether identity impacted the probability of a respondent indicating the pandemic affected their writing using a binomial distribution and logit link function. We then used 'bayesplot' [27] for visualization. For each model, we used weakly informative normal prior distributions (as suggested by experts, e.g., [28]) with a mean of zero and standard deviation of 2.5 and then allowed 'rstanarm' to automatically scale and center predictors and adjust scales of the priors during each run, which are also the default settings in 'rstanarm' [26]. We ran four chains for 10,000 iterations and discarded the first half as warm-up to obtain 20,000 simulations for analysis. We confirmed convergence using the Gelman-Rubin statistic ($R_{hat} < 1.01$) and by examining trace plots. None of the models had influential outliers as assessed by leave-one-out cross-validation ("loo") in the 'rstan' package [29]. We report model coefficients as the median ($\beta_{hat}$ in text, point estimates in Figs 1 and 3) and credible intervals (e.g., a 95% credible interval indicates there is a 95% probability that the true parameter lies within that range). Bayesian posterior distributions are generally more intuitive because they are probabilistic [30], so we used the thousands of iterations per model (i.e., posterior distribution) to look at the probability of the coefficient being positive or negative, which we report in text as the % probability a coefficient is < or > 0.

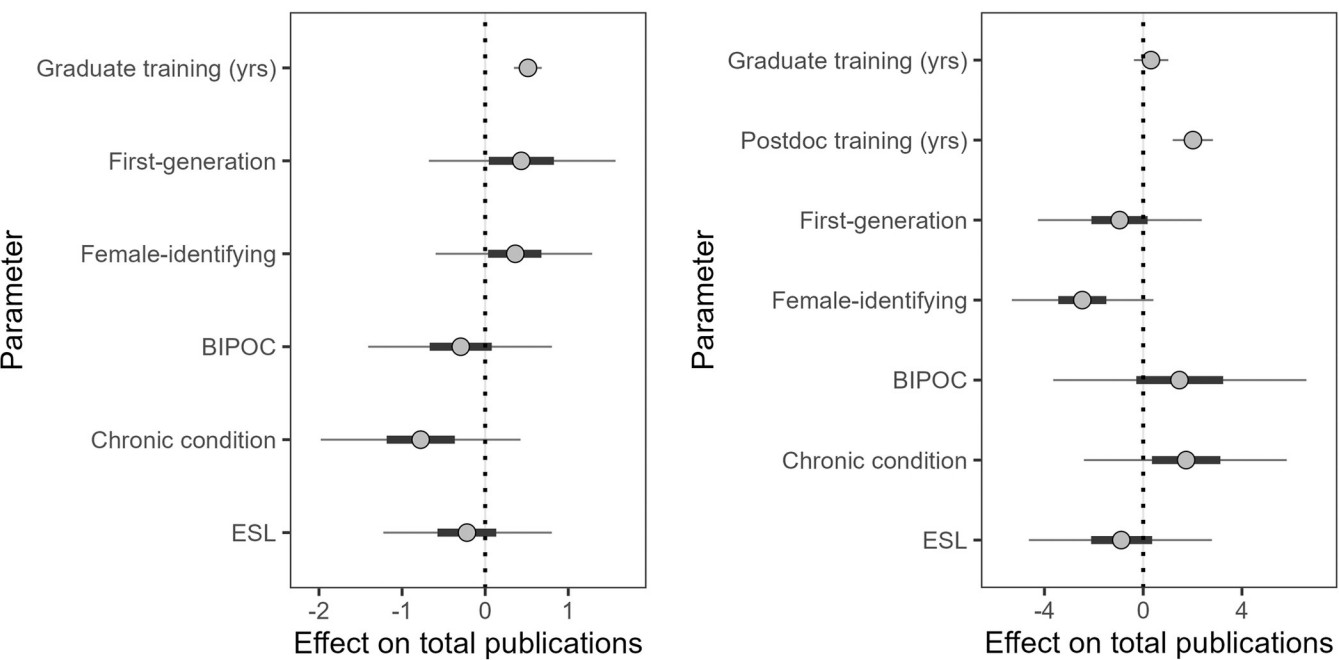

**Fig 1.** The effect of years in training and identity on total publications for A) graduate students (n = 229) and B) postdoctoral scholars (n = 79) leading up to the COVID-19 pandemic. Multiple regression models suggest that identity of graduate students and postdoctoral scholars predicts publication totals. Each point is the median parameter estimate, thick lines are 50% credible intervals (CRIs), and thin lines are 95% CRIs. Graduate and postdoc yrs indicate the number of years in training (as a continuous variable). First gen = first in family to graduate from college; female = female respondents; BIPOC = Black, Indigenous, and/or a person of color; chronic condition = chronic health condition or disability; and ESL = respondents with English as a second language as categorical variables (yes/no).

Our primary response of interest was the total number of trainee publications at time of survey. We have no way of distinguishing which publications were from pre-pandemic vs. during the pandemic, so we assume total publications represents a pre-pandemic metric for two reasons. First, our survey was open less than a year into the pandemic (~11 months since shutdowns began), so the pandemic represented a small fraction of total training time. Secondly, time from full manuscript to publication is often a year or more, so the total number of publications should mainly reflect pre-covid output. We used years as a graduate student and as a postdoc as continuous variables in the models to adjust for baseline differences in publication output due solely to career stage. For example, a first-year graduate student will likely have fewer publications than a third-year postdoc. Identity factors (defined fully in Box 1) were coded as whether the respondent was part of the group (yes = 1) or not (no = 0). Identity factors included whether someone was the first in their family to obtain a college degree (first generation, hereafter first-gen); gender identity; whether a trainee identified as BIPOC; whether the trainee had a chronic health condition or disability (hereafter, chronic condition); and whether the trainee's first language was English or not. Unfortunately, we had too small of a sample size of individuals identifying as non-binary/other (e.g., non-binary, third gender) to get accurate model results (n = 8/311 respondents), so we eliminated all but those who identify as male or female from the analysis on publication totals. Recognizing that these individual survey respondents have shared insights from their experiences, we express deep gratitude for their voluntary contributions. Although these data were too minimal to statistically test how being gender non-binary affected publication outcomes, we encourage readers to view a summary of these responses (S6 Table).

## Results

### Sample frame

We had 342 survey respondents—292 of whom finished the entire survey (85% completion rate)—from 149 subfields of environmental biology (S1 Fig). The most common subfield descriptions included "ecology" (n = 158), "biology" (n = 44), "evolutionary" (n = 34), and "plant" (n = 29). Our sample frame included graduate students from early to advanced stages (0–15 years, avg = 4.7 yrs, SD = 2.7, n = 231/311), and postdocs who had a similar range of experiences from early to advanced stages (0–9 yrs, avg = 1.9, SD = 1.9, n = 80/311).

Respondents represented a diversity of identities (S1 Table). The majority of graduate students (73%, n = 140/193) and postdoc respondents (61%, n = 46/76) self-identified as female (Box 1), while 24% and 37% (n = 47/193 and n = 28/76; graduate students and postdocs, respectively) self-identified as male, and a small number self-identified as non-binary, third gender, or other (3%, n = 6/193; and 3%, n = 2/76). Most respondents (92%) were previously or are currently at a university in the U.S. or Canada; therefore, our results mainly reflect trainee experiences in this region of North America.

Overall, 20% (n = 53/267) of respondents self-identified as BIPOC, comprising 24% of graduate students (n = 46/193) and 9% of postdocs (n = 7/74). First-gen college graduates made up 24% of both graduate student (n = 46/194) and postdoc (n = 19/78) respondents. Additionally, 18% of respondents reported having a disability or chronic health condition (Box 1), with 20% of graduate students (n = 39/193) and 13% of postdocs (n = 10/77). Finally, 18% (n = 36/199) and 23% (n = 18/78) of graduate students and postdocs, respectively, reported that English was their second language (ESL, Box 1).

### Publishing output is predicted by training and identity

A substantial number (40%) of trainees had published prior to starting graduate school (n = 123/310). Postdocs had more first- (avg = 5.1 versus 1.7) and co-authored publications (avg = 5.6 versus 2.5) on average than graduate students and the variation in both metrics was higher among postdocs (S2 Fig). The strongest effect on publication total was the number of years spent as a trainee (Fig 1, S2 and S3 Tables). Each additional year in graduate training resulted in an additional half of a publication for graduate students ($\beta_{hat}$ = 0.52, 100% of posterior samples [hereafter probability] > 0; Fig 1A) and slightly less for postdocs ($\beta_{hat}$ = 0.31, 81% probability > 0; Fig 1B). Years spent as a postdoc had a strong effect. Each additional postdoc year resulted in two more publications ($\beta_{hat}$ = 2.10, 100% probability > 0; Fig 1B).

Our models found contrasting effects of identity on publication totals between graduate students and postdocs (Fig 1, S2 and S3 Tables). First-gen identity was associated with a slight increase in publication totals for graduate students ($\beta_{hat}$ = 0.43, 77% probability > 0), but had little effect on postdocs ($\beta_{hat}$ = -0.96, 71% probability < 0). Female identity was a neutral factor for graduate students ($\beta_{hat}$ = 0.36, 77% probability < 0), but reduced postdoc publication total by 2.5 ($\beta_{hat}$ = -2.47, 95% probability < 0). BIPOC graduate students had slightly fewer publications than non-BIPOC peers ($\beta_{hat}$ = -0.29, 70% probability < 0), but BIPOC postdocs published more papers than non-BIPOC peers ($\beta_{hat}$ = 1.47, 72% probability > 0). Graduate students with chronic conditions had approximately one fewer paper ($\beta_{hat}$ = -0.78, 90% probability < 0), but postdocs with chronic conditions had more papers ($\beta_{hat}$ = 1.74, 80% probability > 0). Having ESL had little effect on publishing productivity of graduate students ($\beta hat$ = -0.22, 66% probability < 0) and postdocs compared to non-ESL peers ($\beta_{hat}$ = -0.89, 68% probability < 0), although ESL trainees were the only identity to provide qualitative descriptions of their difficult experiences in an open response (Table 1).

**Table 1. In the optional open question "*Is there anything else you would like to add about your writing experiences*?" trainees with ESL were the only ones who responded about how their identity affected writing and publishing.** While the model indicated that having ESL was not detrimental to publication output, these stated experiences of trainees with ESL highlighted the challenges.

| Career stage | Gender identity | Comment |
|---|---|---|
| graduate student | male | "As a non-native speaker, the hardest part is to find the right words to properly communicate. But it is rewarding to see a final product, even if it's just a paragraph." |
| postdoc | male | "English writing obligation is unfair" |
| graduate student | female | "[It is] especially hard being a non-native English speaker. Many mental roadblocks and perfectionism make it nearly impossible to be motivated to write. Also, I don't get positive feedback often enough, which makes me feel like I'm not writing well enough and not capable of science." |
| postdoc | male | "Many times I don't know how to express what's in my mind in English. [I spend] a lot of time to find appropriate words for what I want to say." |
| graduate student | female | "The most difficult part for me has been training my brain to think in English. My first language is Spanish and I learned English . . .[at] 23 years old, so always I need a native speaker to check my docs. The [lack of] language diversity in science obligates people to think, write, and speak in English, which makes it difficult. . ." |
| postdoc | none given | "A lot of people have issues due to [having ESL] and it should be taken more into account in Academia." |
| graduate student | female | "I have found writing challenging, especially because English is my second language. . ." |

## Differential impact of the COVID-19 pandemic on perceived writing time, productivity, and motivation

The majority of respondents (70%, n = 197/279) reported that the pandemic impacted their writing habits (Fig 2A). When respondents answered "yes" to the parent question "Has the COVID-19 pandemic impacted your writing habits?" they were directed to a series of nested questions about perceived writing time, productivity, and motivation during the pandemic. Most trainees (52%, n = 101/195) reported more or much more time for writing during the pandemic (Fig 2B), but a combined 75% of respondents (n = 147/196) reported that they felt less or much less productive during the pandemic (Fig 2C). Similarly, 76% (n = 148/196) of respondents reported feeling less or much less motivated to write during the pandemic (Fig 2D).

Participants' reported experience with their writing habits during the pandemic was unequal across identities. Specifically, most female respondents (74%, n = 131/176) reported that the pandemic impacted their writing habits, compared to 61% of those who identify as male (n = 46/75) and 63% (n = 5/8) of non-binary trainees. Strikingly, 85% of BIPOC trainees reported that their writing habits were impacted by the pandemic compared to 67% (n = 142/212) of white peers.

Our models that accounted for identity and years in graduate school supported these results and suggested with a high degree of statistical support (S4 Table) that first-gen, female, or BIPOC graduate students were more likely to have the pandemic impact their writing habits (Fig 3A). First-gen graduate students had a 71% probability and female graduate students had a 77% probability of answering that their writing habits were affected. BIPOC graduate students had the highest (88%) probability of reporting the pandemic affected their writing. Number of years spent in graduate school was unrelated to the impact of the pandemic on writing (1 yr = 51% versus 5 yrs in graduate school = 53% probability). Interestingly, graduate students with ESL were strongly unlikely to report that the pandemic affected writing habits (100% posterior < 0), with only a 25% probability (Fig 3A, S4 Table).

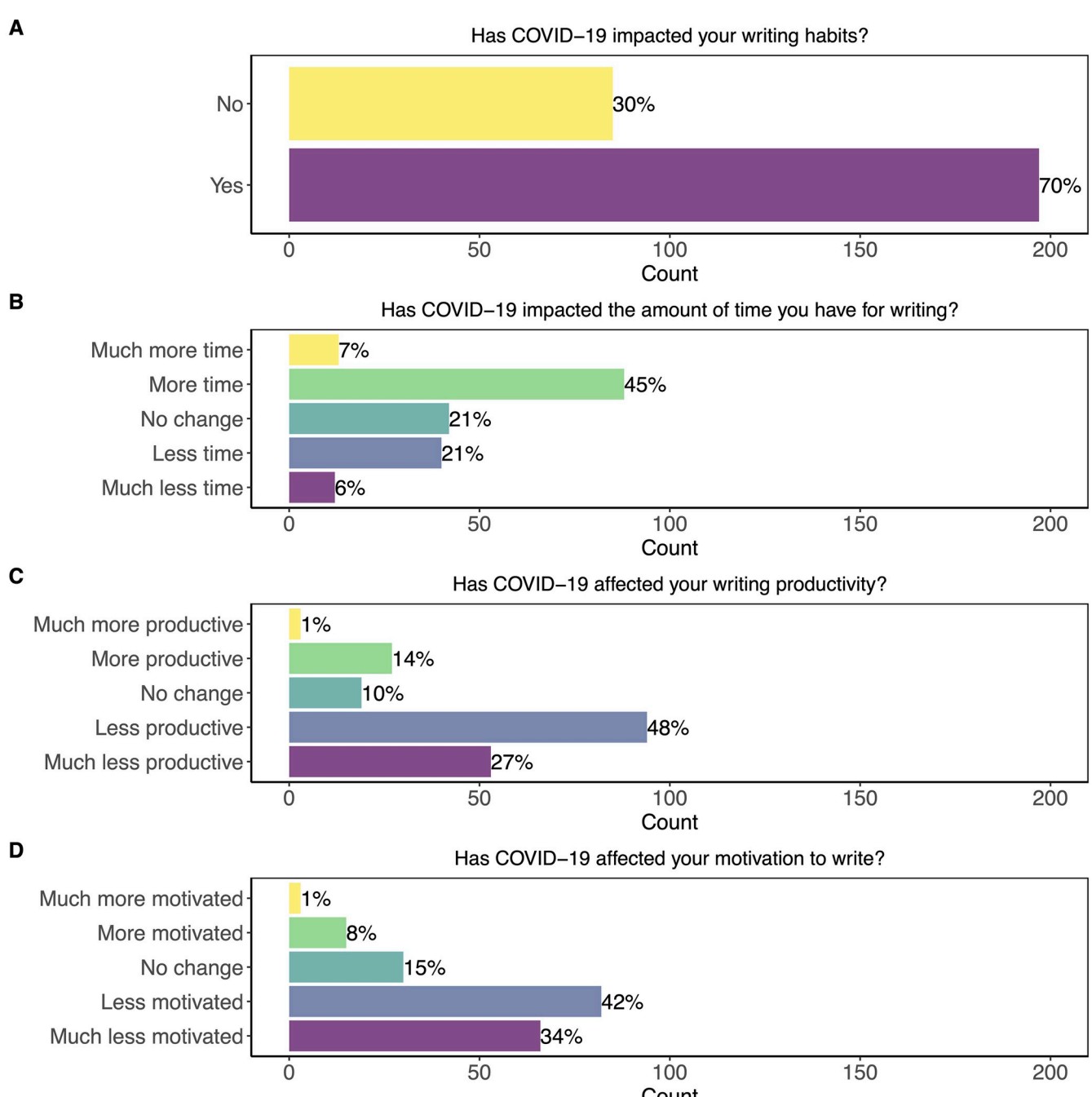

**Fig 2. The impact of the COVID-19 pandemic on trainee writing habits and perceptions of productivity and motivation.** A) When answering the question: "*Has the COVID-19 pandemic impacted your writing habits?*" the majority of respondents said yes. We then asked follow-up questions on perceived effects of the pandemic on time for writing, productivity, and motivation. B) While many respondents reported having more or much more time for writing, C) most respondents reported that they felt less or much less productive during the pandemic. D) Similarly, the majority of respondents reported feeling less or much less motivated.

Among postdocs, those who have spent longer in postdoctoral training and who identify as female were the most likely to report that the pandemic affected their writing, whereas post-docs with chronic conditions and ESL were statistically unlikely to report an effect on writing habits (Fig 3B, S5 Table). Each additional year of postdoc training increased the probability of

## A) Graduate student trainees    ## B) Postdoctoral scholar trainees

**Fig 3.** Binomial multiple regression models suggest identity of graduate students and postdoctoral scholars influenced yes/no responses of A) graduate students (n = 229) and B) postdoctoral scholars (n = 79) to the question "*Has COVID-19 impacted your writing habits?*". All estimates are in logit scale for ease of comparison. More positive values indicate a higher probability of answering "yes." Points are the parameter estimate medians, thick lines are 50% credible intervals (CRIs), and thin lines are 95% CRIs. Graduate and postdoc yrs indicate the number of years in training (as a continuous variable). First gen = first in family to graduate from college; female = female; BIPOC = Black, Indigenous, and/or a person of color; chronic condition = chronic health condition or disability; and ESL = English is the second language of respondents coded categorically (yes/no).

the pandemic affecting writing habits by 6% (96% posteriors > 0; e.g., 1 yr postdoc prob = 57% versus 4th yr prob = 76%). Female postdocs had a 71% probability of being affected by the pandemic (96% posteriors > 0). Postdocs identifying as first-gen and BIPOC reported little to no effect of the pandemic on writing (Fig 3B, S5 Table). Postdocs with ESL had only a 22% probability (95% posteriors < 0) of saying the pandemic affected their writing, and postdocs with chronic conditions a 21% probability (97% posteriors < 0).

## Discussion

We show the pandemic substantially and unequally affected writing habits of trainees. The impacts of the pandemic were disproportionately experienced by HEGs—85% of BIPOC and 74% of female trainees reported that their writing habits were impacted. While our sample frame only included environmental biology fields, the pressure to publish and the inequalities of academia are not exclusive to environmental biology fields. Our results suggest that the pandemic may worsen diversity in academia because its current structures and insufficient support may push historically excluded trainees to leave the academic pipeline. It is essential to acknowledge that basing hiring mainly on trainee publication and grant output disadvantages HEGs and ignores other strengths (e.g., teaching, leadership, mentoring, being a good community member) that academic candidates might offer.

### Diversity, equity, and inclusion in scientific writing

Our results strongly demonstrate that female postdoctoral trainees and graduate students with chronic conditions had lower publication output pre-pandemic compared to their peers

(Fig 1). The emphasis on high publication output in academia makes it likely that lower outputs at some point early in one's career will have a lingering effect regardless of future progress. Interestingly, graduate students and postdocs often had contrasting results with respect to how identity affected publication output. Graduate students had similar publication records across genders, but female postdocs had 1.4 fewer publications than their male counterparts. Postdocs are more likely to have children than graduate students given that they tend to be older [31–33], and this could disproportionately affect the child-carrying and primary caregiving partner, although further research is needed to explore this. However, even for women without children, gender schemas [34] and deliberate and systematic obstacles (i.e., the hostile obstacle course [12]) make it harder for women to succeed in academia. This causes women to leave during the transition from graduate student to postdoc [19] and from postdoc to faculty [35].

First-gen graduate students had more publications than their non-first-gen peers but there was no difference at the postdoctoral level (Fig 1). Others have found first-gen biology doctoral students have similar experiences and outcomes as non-first-gen peers, matching our results [36], but our weak trend at the postdoctoral level shows that we need more data to understand how first-gen postdocs might respond to the pressures of postdoctoral positions and uncertainty in the job market. BIPOC trainees and trainees with chronic conditions tended to have fewer publications as graduate students but more as postdocs (Fig 1). People from HEGs finishing STEM PhDs are half as likely to have submitted a paper in the previous year compared to people from non-HEGs [6] and in-depth surveys suggest high writing anxiety among BIPOC biomedical graduate students [8]. Graduate students experience major institutional sources of stress during their degrees including role strain (e.g., being a teacher but also a student), mentor relationships, isolation, and funding concerns [37]; how these graduate school stressors may build upon the strain of being part of a HEG needs to be explored and addressed.

Trainees with ESL had slightly fewer publications than trainees with English as a first language. Although these trends were weak, many respondents explicitly mentioned the struggles of English as the default language of science in the open response section (Table 1). Research shows that trainees with ESL face enormous obstacles to scientific publishing [38] but writing intervention programs have helped this group gain confidence and skills in scientific writing [8]. Universities should invest in extra resources for trainees with ESL, including discipline-specific writing courses [39,40].

Our results, alongside previous work [6], demonstrate that certain groups of trainees had lower publication outputs than their peers leading up to the pandemic. Academia is currently structured to differentially impact identity groups under baseline conditions [15,22]. The pandemic has highlighted and, in some cases, accentuated the unjust differences that exist for some community members. Although significant and global disturbances like the pandemic are uncommon, the productivity disruptions mirror more singular events that individuals experience such as grief, illness, or unpredictable childcare, so equitably adjusting academic metrics of success will have long-reaching positive effects on future trainees seeking tenure-track positions.

## COVID-19 pandemic and identity

Graduate students and postdocs have navigated the impacts of the pandemic at a critical time in their careers when publishing matters most for their future career success. While recent studies have examined the effect of the pandemic on early-career faculty [24,25,41], our study is the first to our knowledge that quantitatively examines the impact of the pandemic on trainees (but see Suart et al. [42] for a qualitative overview). The differential writing outputs across

trainees may be exacerbated by the ongoing pandemic. The majority of trainees reported that the pandemic has affected their writing habits, with most also feeling less productive and motivated to write (Fig 2). Thus, while these trainees felt that they had more time for writing during the pandemic, it was not actually conducive to writing productivity or motivation. This finding is not surprising; other studies have explored the connection between negative mental health, feelings of apathy, and lower productivity and motivation [42]. Mental health was already a concern for trainees [43], and the pandemic will likely worsen anxiety and depression [44,45]. Furthermore, 70% of respondents preferred working in non-home environments that were unavailable during lockdowns. Many trainees were likely adjusting to working from home while living in smaller shared spaces with roommates or family. Trainees who are parents had the additional challenge of caring for their children, and women in particular have been disproportionately affected by increased childcare or other family caretaking responsibilities stemming from the pandemic [24,25,46,47].

However, the pandemic has not affected all trainees equally. While there was no effect of years spent in graduate school, each additional year of postdoctoral work strongly increased the reported effect of pandemic on writing habits (Fig 3). Most postdoc positions in environmental biology fields are for 2–3 years at most, so losing 1.5 years of access to networking, lab work, and field sites was devastating, especially when paired with grim job prospects [48,49] and the importance of publishing during postdoctoral work [2]. Despite calls for more support and contract extensions [50,51], many universities did not accommodate postdocs. Postdocs are deeply concerned that the pandemic has worsened their overall career prospects [51,52].

We also found evidence that female trainees were disproportionately affected. The experiences of non-binary and other gender trainees were similar (S5 Table), but more data are needed to more fully characterize their experience throughout the pandemic. Other studies have also reported gender inequalities in research productivity during the pandemic, including women having less time for research [25] and being underrepresented in COVID-19-related research and authorship [46,53,54]. Actions such as creating niche funding opportunities to support women and gender minorities and developing flexible working schedules for those with childcare responsibilities have been proposed as strategies for supporting these early-career scientists [24]. Additionally, creating institutional resources for writing (e.g., courses, workshops, support groups) could help mitigate the effects of unexpected disturbances.

Disparities in the pandemic's impact on writing were most pronounced for BIPOC respondents. A striking 85% of BIPOC trainees reported that the pandemic affected their writing habits, and overwhelmingly felt unproductive and unmotivated to write. The timing of the pandemic also coincided with the collective trauma felt by Black Americans in response to police brutality and the murders of Breonna Taylor and George Floyd, among others. Furthermore, violence directed at Asian Americans has been on the rise in the U.S. during the pandemic [55]. We recognize that this violence transcends workspaces and can manifest in field work settings as well [15,56]. Although our data do not differentiate between the public health and social justice crises, the compounding issues of the pandemic, racism, and the rise of racially-charged violence likely contributed to the decreased writing motivation felt by BIPOC trainees. It is critical for BIPOC trainees to take care of their mental health [57], and for academic communities to support and retain BIPOC trainees by dismantling institutional white supremacy and creating inclusive environments where BIPOC scholarly excellence is celebrated [15,21,22,58]. Future research should aim to disentangle the role of longstanding inequities in academia and the effects of recent events (e.g., the pandemic and the social justice movement of 2020) on work output.

First-gen graduate students were also more likely to report a disruption to writing during the pandemic, which may be due to the mental toll of having more family and friends with low

socioeconomic status who struggled more throughout the pandemic [59]. Surprisingly, trainees with ESL and those with chronic conditions were statistically less likely to report the pandemic affected their writing habits (Fig 3). In the case of trainees with ESL— many of whom are likely international trainees— the pandemic prevented many from being able to see their families. However, these trainees already were physically distanced from their families even prior to the pandemic, which could explain why they were less likely to report the pandemic affected their writing habits given that they could have already been experiencing isolation. More research is needed to understand why these trainees reported less interruption to writing habits.

## Limitations of the study

Given differences in publishing and research between STEM fields, our results most prominently reflect the experiences of environmental biology trainees. While our findings can inform our understanding of the pandemic's impact, future research is needed to explicitly explore the effects of the pandemic on trainees across a wider range of identities and STEM fields. We chose an anonymous survey mechanism to maximize and standardize our sample size, but in doing so, we lack the nuance we could have included if our study had also allowed for follow-up interviews. The survey questions were subject to interpretation and reporting differences among respondents. Because of this, our analyses had to be neutral; for example, we asked trainees whether the pandemic impacted their writing, however we cannot fully conclude it was a negative impact, despite follow-up questions showing a perceived lack of motivation and productivity. Additionally, although ESL trainees did not report effects of the pandemic on writing habits, it could very well be that cultural stigmas around discussing difficulties prevented them from answering our questions honestly, especially considering the free responses indicated writing in general was much more challenging for ESL trainees (Table 1). The survey also lacked the ability to differentiate between ESL trainees who reside in their home country and international students. Furthermore, caretaker status would have been an informative metric due to school and care center closure during the pandemic, as this likely had a very strong influence on productivity. Future research should disentangle these effects further.

The intersection of identities, such as race and gender, is important [17,60] and undoubtedly would also add new insight and a fine-tuned understanding of how different identities were affected by the pandemic. To adequately explore intersectionality, we would have had to specifically target responses from people at the intersection of the identities of interest. As presented here, the five aspects of identity in our survey (race, gender, first-gen, ESL, and chronic conditions/disability) could be assessed additively. Although this does not quantify interactive effects, the sum of identities provides the minimum effect one could expect on publishing output. We encourage future studies to quantitatively assess intersectionality of different identities and include more nuance in their assessments.

## Equity in academia

We show that the pandemic disproportionately affected historically excluded environmental biology trainees. Female, BIPOC, and first-gen trainees were much more likely to indicate that the pandemic impacted their writing habits. These writing interruptions may linger on the CVs of those who were trainees during the pandemic, manifesting as disparities in publication counts compared to more productive colleagues and competitors. To recover from the pandemic, we should reconsider how to evaluate job candidates in future hiring. The current academic evaluation metrics are not equitable because they do not (1) account for the different

input into individual publications, (2) give equal weight to the non-publishing aspects of academia (e.g., teaching, mentoring, JEDI), or (3) account for the means by which academics achieve their outputs [15]. Research projects that are theory-, lab-, or field-based require different inputs from inception to publication. Furthermore, some scientists may encounter hostile situations during field work that can delay progress [56]. Search committees should consider these project inputs when evaluating candidates beyond total publication counts and journal impact factors.

Thus, while peer-reviewed publications maintain the quality and knowledge-base of science, the non-publishing aspects of academia should be given more weight in the evaluation process. For example, on its own, a candidate's publication output does little to impact the issue of undergraduates from HEGs leaving STEM degrees [61] or graduate students leaving academia. However, a candidate with training and experience in inclusive teaching practices and active participation in JEDI initiatives is likely going to improve their prospective institution by being both a good instructor and mentor. Finally, while a candidate can have an exceptional track record of publications and grants, they may have achieved those outputs while harming community members around them [15]. The differential impacts of the pandemic on HEGs, along with the call to create and retain a diverse and equitable faculty is an opportunity for universities to reevaluate and adapt their current metrics. For example, non-white, non-male, and first-gen faculty are more likely to participate in JEDI activities [62]; if trainees engaging in and prioritizing institutional service and JEDI initiatives have similar identities, these individuals may have less time to dedicate to writing. To diversify academia, including non-publishing aspects of a candidate as measures of success would help create a more equitable evaluation process during hiring.

## Conclusions

Given our results showed that baseline publications and perceptions of writing during the pandemic differed among identities, the impacts of the pandemic could linger on the CVs of many trainees as they seek academic positions. Therefore, we encourage committees to evaluate faculty candidates not only on scientific contributions (e.g., publications), but also consider how individuals will serve as advisors and community members. Currently in academia there is an overemphasis on publication totals as an almost singular metric of scientific success [3], and publication totals alone do not account for inequities among groups (this study), account for the means by which those metrics were achieved [15,63], or adequately support HEGs [21,57]. Comprehensively assessing trainee contributions by including service, teaching, mentorship, and JEDI initiatives would make academia more inclusive, vibrant [64], and resilient–especially post-COVID-19.

## Supporting information

**S1 Fig. Word cloud representation of respondents' subfield of environmental biology.** Bigger words indicate higher occurrence in the survey responses.
(PDF)

**S2 Fig.** Smoothed density of A) first-authored and B) co-authored publications of respondents separated by career stage. Career stage grad represents graduate students, while postdoc represents postdoctoral scholars.
(PDF)

**S1 Table. Breakdown of survey respondent identity.** We allowed respondents to choose their identity preference or NA if they were uncomfortable answering. Below the results are

reported as a percentage of total responses within each career stage.
(PDF)

**S2 Table. Graduate student results for Bayesian multiple regression of how years in school and identity affect publication output.** Except for years in graduate school, all other variables are factorial and coded as 1 = trainee identifies or 0 = trainee does not identify as first generation, female, BIPOC, having a chronic condition, or having English as a second language (ESL). Variables with 80% or higher probability of being on the same side of zero as the estimate (PD sign match) are bolded. 95% CRI = 95% credible interval, ESS = effective sample size.
(PDF)

**S3 Table. Postdoctoral scholar model results for Bayesian multiple regression of how years in training and identity affect publication output.** Years spent as a graduate student and postdoctoral scholar are continuous, and all other variables are factorial and coded as 1 = trainee identifies or 0 = trainee does not identify as first generation, female, BIPOC, having a chronic condition, or having English as a second language (ESL). Variables with 80% or higher probability of being on the same side of zero as the estimate (PD sign match) are bolded. 95% CRI = 95% credible interval, ESS = effective sample size.
(PDF)

**S4 Table. Results for binomial multiple regression on yes/no responses among graduate students to the question "*Has COVID-19 impacted your writing habits*?".** All estimates are in logit scale for ease of comparison. Variables with 80% or higher probability of being on the same side of zero as the estimate (PD sign match) are bolded.
(PDF)

**S5 Table. Binomial multiple regression on yes/no responses among postdoctoral scholars to the question "*Has COVID-19 impacted your writing habits*?".** All estimates are in logit scale for ease of comparison. Variables with 85% or higher probability of being on the same side of zero as the estimate (PD sign match) are bolded.
(PDF)

**S6 Table. Five of eight non-binary/other gendered respondents indicated that the COVID-19 pandemic impacted writing habits.** They generally reported having less productivity and less motivation.
(PDF)

**S1 File. A full copy of Qualtrics survey used to collect our data.** This copy shows exact question phrasing and survey structure.
(PDF)

## Acknowledgments

Thank you to the trainees that answered our call for survey responses about writing. We would not have any data without you. We also thank the Yale StatLab at the Marx Science and Social Science Library for feedback on survey design. J. Monk, N. Harris, S. Gámez, L. Baik, K. McConnell, C. Wilkinson, and M. McCary provided helpful feedback on earlier versions of the manuscript.

## Author Contributions

**Conceptualization:** Freya E. Rowland, Kyra A. Prats, Yara A. Alshwairikh, Mary K. Burak, Ana Clara Fanton, Marlyse C. Duguid.

**Data curation:** Freya E. Rowland.

**Formal analysis:** Freya E. Rowland, Kyra A. Prats.

**Investigation:** Freya E. Rowland, Kyra A. Prats, Yara A. Alshwairikh, Mary K. Burak, Ana Clara Fanton.

**Methodology:** Freya E. Rowland, Kyra A. Prats, Yara A. Alshwairikh.

**Project administration:** Marlyse C. Duguid.

**Visualization:** Freya E. Rowland, Kyra A. Prats, Ana Clara Fanton.

**Writing – original draft:** Freya E. Rowland, Kyra A. Prats, Yara A. Alshwairikh, Mary K. Burak, Ana Clara Fanton.

**Writing – review & editing:** Freya E. Rowland, Kyra A. Prats, Yara A. Alshwairikh, Mary K. Burak, Ana Clara Fanton, Marlyse C. Duguid.

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
