## [Decision Letter · Decision Letter 0]

23 Nov 2022

PONE-D-22-24997

Overemphasis on publications may disadvantage historically excluded groups in academia before and during COVID-19

PLOS ONE

Dear Dr. Rowland,

Thank you for submitting your manuscript to PLOS ONE. After careful consideration, we feel that it has merit but does not fully meet PLOS ONE’s publication criteria as it currently stands. Therefore, we invite you to submit a revised version of the manuscript that addresses the points raised during the review process.

We look forward to receiving your revised manuscript.

Kind regards,

Maria Elisabeth Johanna Zalm, Ph.D

Editorial Office

PLOS ONE

Journal Requirements:

2. For reproducibility purposes please include a copy of the survey used in this analysis.

 "The authors received no specific funding for this work. FER was supported on a YIBS Donnelley Postdoctoral Fellowship. "

7. Please remove your figures from within your manuscript file, leaving only the individual TIFF/EPS image files, uploaded separately. These will be automatically included in the reviewers’ PDF.

Additional Editor Comments:

Your manuscript has been assessed by two peer-reviewers and their reports are appended below.

The reviewers comment that the current version of the manuscript does not contain sufficient information to carry out a detailed assessment of the study, and that a copy of the questionnaire is required for a detailed article assessment. In addition, the reviewers comment that aspects of the study require additional detail and/or clarification. 

Could you please carefully revise the manuscript to address all comments raised?

Reviewers' comments:

Reviewer's Responses to Questions

**Comments to the Author**

1. Is the manuscript technically sound, and do the data support the conclusions?

Reviewer #1: Yes

Reviewer #2: Partly

2. Has the statistical analysis been performed appropriately and rigorously? 

Reviewer #1: Yes

Reviewer #2: Yes

3. Have the authors made all data underlying the findings in their manuscript fully available?

Reviewer #1: Yes

Reviewer #2: No

4. Is the manuscript presented in an intelligible fashion and written in standard English?

Reviewer #1: Yes

Reviewer #2: Yes

5. Review Comments to the Author

Reviewer #1: This article was well-written and supported by the data presented in the manuscript. The authors ask a very important question as to how those members of historically excluded groups were affected by the COVID-19 pandemic. They show that grad students and post-docs are affected differently, but those who identify as BIPOC were disproportionately affected by the COVID-19 pandemic. They illustrated nicely that these effects will be far-reaching as it will affect the hiring of BIPOC and individuals from other historically excluded groups and implored academia to consider other measures of success when it comes to the quality of job candidates. I think the authors did a fantastic job with this paper and I cannot wait to see it in print.

Reviewer #2: The article aims to assess how identity interacts with scholarly publishing and how the COVID-19 pandemic influenced feelings of scholarly writing productivity based on identity. The authors rely on an online survey conducted with graduate and post-doctoral students in environmental biology (n=342). The authors find that female graduate students and those with chronic conditions lag in publication output as compared to other groups. The authors found mixed evidence for BIPOC and first-generation college students. I applaud the authors for mainly focusing the project on trainees, their attempt to untangle the effects of identity, pre-pandemic productivity and highlighting possible post-pandemic effects. I have few comments for the authors that I hope they will find them helpful.

• The authors didn’t provide a copy of the questionnaire, so it is hard to see what questions were included and how they were worded.

• Related to the first point, it is unclear how the authors measured pre-pandemic productivity and how they built this assumption/relationship between years of training and number of publications.

• I am not also sure about the value of including the results of the respondents’ reported productivity prior to the pandemic and how do they define publications (peer-reviewed, scientific reports, blog post…ect). Again, since the survey instrument is not provided, it is hard to make sense of these findings.

• What is the response rate of the survey?

• Definition of identity and how they conceptualize identity is currently in the appendix, but I would encourage the authors to include in the paper and be clear upfront what they mean by identity and how they are using this concept in the paper. I am also not sure how/why speaking English as a second language is an identity marker in this paper.

• Another main issue with this paper is the question the authors use to build their analyses on post-pandemic productivity. The authors use the question: ““Has the COVID-19 pandemic impacted your writing habits?” The wording of this question is so vague to use as the main variable to measure productivity across groups as used in Figure 3. Especially that only 27 percent of your sample reported having less time to spend on writing (Figure 2B).

• The authors’ most consistent findings are those related to gender identity which is consistent with previous work and their n is currently too small to tell us any information about intersectional identities (BIPOC and gender for instance) which I believe would be the main strength of this project.

• Going back to the gender findings, they don’t really tell us much about why this is the case since they are not reporting any information on the marital or parenthood status of the females in their sample. I find these findings rather incomplete without this information.

6. PLOS authors have the option to publish the peer review history of their article (what does this mean?). If published, this will include your full peer review and any attached files.

Reviewer #1: No

Reviewer #2: No

---

## [Decision Letter · Decision Letter 1]

21 Mar 2023

PONE-D-22-24997R1Overemphasis on publications may disadvantage historically excluded groups in academia before and during COVID-19PLOS ONE

Dear Dr. Rowland,

Thank you for submitting your manuscript to PLOS ONE. After careful consideration, we feel that it has merit but does not fully meet PLOS ONE’s publication criteria as it currently stands. Therefore, we invite you to submit a revised version of the manuscript that addresses the points raised during the review process.

 Revise submission provide adequate changes to reflect most of the reviewer comments. Now the presentation provide easy to follow strucutre. However, based on R3 comments on revised manuscript, it is important to address those 6 points in the paper  - specially I look forward to seeing the clarifications explained about methodological missing parts, and results section to provide evidence to your answer in research questions. Although there is a comment about the tittle of the paper, I would encourage authors to re-think bit more and change of you belive the title does not reflect exactly the research 

We look forward to receiving your revised manuscript.

Kind regards,

Dilrukshi Gamage, Ph.D

Academic Editor

PLOS ONE

Journal Requirements:

Additional Editor Comments :

Thank you authors for the revisions and the rebuttal letter.

After reading the revise manuscript, reviewers are convinced that the paper has fulfilled its goals and presented adquately. However, Reviewer 3 has some very valid questions that authors need to address when they explain the paper.

Specially towards the 6 points explained By the R3. I would like the authors to adress them specialy the abstract to be succinently provide related evidence and results / discussion section. You may leave the title as it is if you think it more suitable but rest of the points ( 6 all together) need to be addressed in the paper.

Reviewers' comments:

Reviewer's Responses to Questions

**Comments to the Author**

1. If the authors have adequately addressed your comments raised in a previous round of review and you feel that this manuscript is now acceptable for publication, you may indicate that here to bypass the “Comments to the Author” section, enter your conflict of interest statement in the “Confidential to Editor” section, and submit your "Accept" recommendation.

Reviewer #1: All comments have been addressed

Reviewer #3: (No Response)

2. Is the manuscript technically sound, and do the data support the conclusions?

Reviewer #1: Yes

Reviewer #3: No

3. Has the statistical analysis been performed appropriately and rigorously? 

Reviewer #1: Yes

Reviewer #3: N/A

4. Have the authors made all data underlying the findings in their manuscript fully available?

Reviewer #1: Yes

Reviewer #3: No

5. Is the manuscript presented in an intelligible fashion and written in standard English?

Reviewer #1: Yes

Reviewer #3: Yes

6. Review Comments to the Author

Reviewer #1: This article was well-written and supported by the data presented in the manuscript. The authors ask a very important question as to how those members of historically excluded groups were affected by the COVID-19 pandemic.They show that grad students and post-docs are affected differently, but those who identify as BIPOC were disproportionately affected by the COVID-19 pandemic. They illustrated nicely that these effects will be far-reaching as it will affect the hiring of BIPOC and individuals from other historically excluded groups and implored academia to consider other measures of success when it comes to the quality of job candidates. I think the authors did a fantastic job with this paper and I cannot wait to see it in print.

Reviewer #3: 1. title:

Add study design in the title: a survey-based descriptive study

Recommend a more lucid title: Publication productivity of historically excluded trainees of the environmental biology field in the United States and Canada in publication during COVID-19 pandemic: a survey-based descriptive study

The results do not provide the evidence of disadvantage of the historically excluded trainees.

Therefore, the title should be relevant to the content of this research appropriately.

2. Abstract

2-1.BIPOC should be written as a full name at the first appearance.

2-2. historically excluded trainees should be defined at the first part of abstract.

2-3.There is no lucid objectives and methods of this study. "We surveyed 342 environmental biology trainees to assess: (1) how identity interacts with scholarly publishing and (2) how the COVID-19 pandemic influenced feelings of scholarly writing productivity based on identity" is a mention of the methods section. However, there is no more concrete methodsm including source of survey and institute of participants.

2-4. Conclusion - "We urge the academy to consider how an overemphasis on publication output during hiring may affect historically excluded groups—especially in a post-COVID-19 era—and to prioritize accountability for the means by which our outputs are achieved." is not supported by results. What are evidence of overemphasis on publication in this study? Evidence is not found.

3. Introduction

3-1. "Students from underrepresented backgrounds" should be defined lucidly.

4. Materials and methods

4-1. Survey design and distribution: Provide the validity test results of an "anonymous Qualtrics survey"

4-2. Provide the raw response data from participants.

4-3. "chronic condition" is better to be written as chronic diseases of disablities.

4-3. There was no survey before pandemic. Therefore, in this manuscript, before pandemic better to be deleted.

5. Results

5-1. The footnote in Fig. 2 "A) The majority of respondents said that the pandemic

234 has affected their overall writing habits. B) While many respondents reported having

235 more or much more time for writing, C) most respondents reported that they felt less or

236 much less productive during the pandemic. D) Similarly, the majority of respondents

237 reported feeling less or much less motivated." is redundant. Reomve it.

6. Discussion

6-1. Our results strongly demonstrate that female trainees and those with chronic conditions had

289 lower publication output pre-pandemic compared to their peers

 It is not logical to mention this sentence. According to Fig. 1, the publication output is different by group (graduate students and postdoc)

There should be more clear description for interpreting results.

6-2.

the pandemic impacted their writing habits It is uncertain that impact is positive or negative.

6-3.

"To conclude, committees could evaluate faculty candidates not only on scientific

430 contributions (e.g., publications stemming from research grants), but also seek well-rounded

431 individuals who will serve as good advisors and community members.

...

Comprehensively assessing trainee contributions by including service, teaching, mentorship, and

436 JEDI initiatives would make academia more inclusive, vibrant [60], and resilient"

This conclusion can be said without present work. It is too general description. The conclusion should be a

answer to research questions.

7. PLOS authors have the option to publish the peer review history of their article (what does this mean?). If published, this will include your full peer review and any attached files.

Reviewer #1: No

Reviewer #3: No

---

## [Author Response · Author response to Decision Letter 1]

21 Apr 2023

Dear Dr. Gamage,

We are pleased to submit a revision to our publication “Overemphasis on publications may disadvantage historically excluded groups in academia before and during COVID-19” (ONE-D-22-24997R1). We have responded to all reviewer concerns and detail them more below. The comments helped clarify and strengthen the paper.

We look forward to having this work (hopefully) published in PLOS ONE soon. We think it will be a well-cited and received paper.

Best regards,

Freya Rowland and Kyra Prats, on behalf of all coauthors

-----

6. Review Comments to the Author

Reviewer #1: This article was well-written and supported by the data presented in the manuscript. The authors ask a very important question as to how those members of historically excluded groups were affected by the COVID-19 pandemic. They show that grad students and post-docs are affected differently, but those who identify as BIPOC were disproportionately affected by the COVID-19 pandemic. They illustrated nicely that these effects will be far-reaching as it will affect the hiring of BIPOC and individuals from other historically excluded groups and implored academia to consider other measures of success when it comes to the quality of job candidates. I think the authors did a fantastic job with this paper and I cannot wait to see it in print.

 Thank you for your kind assessment of our paper.

Reviewer #3: 1. title:

Add study design in the title: a survey-based descriptive study

Recommend a more lucid title: Publication productivity of historically excluded trainees of the environmental biology field in the United States and Canada in publication during COVID-19 pandemic: a survey-based descriptive study

The results do not provide the evidence of disadvantage of the historically excluded trainees.

Therefore, the title should be relevant to the content of this research appropriately.

We disagree that our study does not present any disadvantage of historically excluded trainees. Our data show lower publication output among trainees from historically excluded groups in STEM. This directly affects the ability of these trainees to secure employment, given the overemphasis on publications in academia (Refs 1-4). Therefore, we think that our title adequately reflects our study.

PLOS limits titles to 150 characters, but we do agree that a better description of our study in the title is warranted. We have edit it to:

Overemphasis on publications may disadvantage historically excluded groups in STEM before and during COVID-19: a survey-based study

Short title: Identity and COVID impacts on trainee productivity

2. Abstract

2-1.BIPOC should be written as a full name at the first appearance. 

 Fixed.

2-2. historically excluded trainees should be defined at the first part of abstract.

We have added the definition of “historically excluded groups in STEM” to our abstract. It now reads: “Our results suggest that the disproportionate impact of the pandemic on writing output may only heighten the negative effects commonly reported amongst historically excluded trainees–i.e., people who have been excluded from full rights based on historical systems of oppression.”

2-3.There is no lucid objectives and methods of this study. "We surveyed 342 environmental biology trainees to assess: (1) how identity interacts with scholarly publishing and (2) how the COVID-19 pandemic influenced feelings of scholarly writing productivity based on identity" is a mention of the methods section. However, there is no more concrete methodsm including source of survey and institute of participants.

We have edited this to include some more details about our respondents. It now reads: “We analyzed anonymous survey responses from 342 North American environmental biology graduate students and postdoctoral scholars (hereafter trainees) about scientific writing experiences of trainees to assess: (1) how identity interacts with scholarly publishing and (2) how the COVID-19 pandemic influenced trainee perceptions of scholarly writing productivity and whether there were differences among identities”

It is difficult to add more detail in the abstract without going over word limits. In our abstract we wanted to mainly highlight our results. We are uncertain wha the reviewer means by the source of information, but have added it was a survey. We don’t have any information about the institutions of respondents. 

We disagree that these are not objectives. We lay out two very specific objectives but have added some detail to make them clearer.

2-4. Conclusion - "We urge the academy to consider how an overemphasis on publication output during hiring may affect historically excluded groups—especially in a post-COVID-19 era—and to prioritize accountability for the means by which our outputs are achieved." is not supported by results. What are evidence of overemphasis on publication in this study? Evidence is not found.

Here we are relating our study to its broader context, which is important. We cite papers throughout that talk about how publication output is the primary metric for hiring (Refs 1-4). We want our study to be a call to reconsider how academia evaluates applicants. 

However, we have edited this concluding sentence: “Based on our findings, we propose that the academy should consider how an overemphasis on publication output during hiring may affect historically excluded groups in STEM—especially in a post-COVID-19 era.”

3. Introduction

3-1. "Students from underrepresented backgrounds" should be defined lucidly.

We changed this to the term we use throughout (“historically excluded groups [HEGs])” and added a reference to Box 1 where we define it.

4. Materials and methods

4-1. Survey design and distribution: Provide the validity test results of an "anonymous Qualtrics survey"

We assume the reviewer is asking about validity as defined in the 2014 Standards for Educational and Psychological Testing as “validation can be viewed as a process of constructing and evaluating arguments for and against the intended interpretation of test scores and their relevance to the proposed use (AERA, APA, NCME, 2014, pg 11).

Our survey sprung from our writing support group and our experiences learning how to write scientifically and navigate the pandemic as trainees. All authors were involved in survey design, we solicited feedback from the Yale Statistics Lab on survey design and sent an initial “test” survey to three colleagues to get feedback on areas that might be confusing. 

Because this was a voluntary, anonymous survey, we are unable to follow-up on how survey respondents may have interpreted questions. Because of this, we are careful in our analyses – e.g., we asked trainees whether the pandemic impacted their writing, however since we never asked if it *negatively* impacted their writing we purposely do not discuss it in this matter.

Are there areas we wish we could go back in time and improve? Absolutely. But the survey was thoroughly vetted for how we could interpret results, length, and accessibility of questions (through the Qualtrics system). We had 10 people examine the survey carefully before we released it for broader distribution and think the results have an interesting perspective to offer on the intersection of identity and the pandemic for trainees. Furthermore, we had a rather large sample size to make inference from statistics. 

4-2. Provide the raw response data from participants.

We have released our data on Dryad where it is freely available for download. Furthermore, our GitHub repository with all associated code and data is public. We will archive the repo on Zenodo upon acceptance. 

We have added the following data availability statement to the manuscript: “The data that support the findings of this study are openly available in Dryad at https://doi.org/10.5061/dryad.fqz612jwn. All associated code is publicly available on our GitHub repository at https://github.com/anacfanton/covid-identity and will be archived on Zenodo for permanent access upon acceptance.” 

4-3. "chronic condition" is better to be written as chronic diseases or disabilities.

We asked “Do you have a disability or chronic health condition?” to encompass a broad range of chronic diseases and disabilities and be as inclusive as possible. We have defined this in the box. We use “chronic condition” in the figures and manuscript as a shorter way to define it for easy of figure interpretation and added “… whether a trainee had a chronic health condition or disability (hereafter, chronic condition)” to the Methods.

4-3. There was no survey before pandemic. Therefore, in this manuscript, before pandemic better to be deleted.

We disagree. Our survey was open from March - April 2021, so less than a year into the pandemic. Considering that publishing timelines are often a year or more (e.g., four months for first review, two months to revise, four months for a second review = 10 months at most ambitious but likely much longer with rejections), the total number of publications should mainly reflect pre-covid output. This will likely be confusing to other readers so we added an explanation to the methods:

“Our primary response of interest was the total number of publications a trainee at time of survey. We have no way of distinguishing which publications were from pre-pandemic vs. during the pandemic, so we assume total publications represents a pre-pandemic metric for two reasons. First, our survey was open less than a year into the pandemic (~11 months since shut-downs began), so the pandemic represented a small fraction of total training time. Secondly, time from full manuscript to publication online is often a year or more, so the total number of publications should mainly reflect pre-covid output.” 

In contrast, the COVID-19 component specifically asked about the impact of the pandemic on writing habits. We focus on trainees' perceived productivity and motivation during the pandemic.

5. Results

5-1. The footnote in Fig. 2 "A) The majority of respondents said that the pandemic

234 has affected their overall writing habits. B) While many respondents reported having

235 more or much more time for writing, C) most respondents reported that they felt less or

236 much less productive during the pandemic. D) Similarly, the majority of respondents

237 reported feeling less or much less motivated." is redundant. Remove it.

(A) is the overarching question about whether the pandemic affected writing habits. (B), (C), and (D) follow up with specific, different aspects of writing. (B) focuses on time for writing, (C) on perceived productivity, and (D) on motivation. These are not the same and therefore we disagree that they are redundant. Perhaps this wasn’t clear in the caption, so we have edited it. 

The caption for Fig. 2 now reads: “The impact of the COVID-19 pandemic on trainee writing habits and perceptions of productivity and motivation. A) When answering the question: “Has the COVID-19 pandemic impacted your writing habits?” the majority of respondents said yes. We then asked follow-up questions on perceived effects of the pandemic on time for writing, productivity, and motivation. B) While many respondents reported having more or much more time for writing, C) most respondents reported that they felt less or much less productive during the pandemic. D) Similarly, the majority of respondents reported feeling less or much less motivated.”

6. Discussion

6-1. Our results strongly demonstrate that female trainees and those with chronic conditions had

289 lower publication output pre-pandemic compared to their peers

 It is not logical to mention this sentence. According to Fig. 1, the publication output is different by group (graduate students and postdoc)

There should be more clear description for interpreting results

Changed to “female postdoctoral trainees” and “graduate students with chronic conditions”

6-2.

the pandemic impacted their writing habits It is uncertain that impact is positive or negative.

One could interpret from Figure 2 that the impact was negative (e.g., less productivity, less motivation), but in the survey we did not ask specifically whether the impact was positive or negative. To be rigorous, we avoid assigning the impacts as positive or negative. 

However, in the discussion we do state “The majority of trainees reported that the pandemic has affected their writing habits, with most also feeling less productive and motivated to write (Fig 2). Thus, while these trainees felt that they had more time for writing during the pandemic, it was not actually conducive to writing productivity or motivation.”

6-3.

"To conclude, committees could evaluate faculty candidates not only on scientific

430 contributions (e.g., publications stemming from research grants), but also seek well-rounded

431 individuals who will serve as good advisors and community members.

...

Comprehensively assessing trainee contributions by including service, teaching, mentorship, and

436 JEDI initiatives would make academia more inclusive, vibrant [60], and resilient"

This conclusion can be said without present work. It is too general description. The conclusion should be an answer to research questions.

We have reworded the concluding paragraph (and added a header) to more strongly tie back in our results. Similar to our response above, we want to reiterate in our conclusion that we hope our results are a call to action in academia. 

7. PLOS authors have the option to publish the peer review history of their article (what does this mean?). If published, this will include your full peer review and any attached files.

Do you want your identity to be public for this peer review? For information about this choice, including consent withdrawal, please see our Privacy Policy.

Reviewer #1: No

Reviewer #3: No

---

## [Decision Letter · Decision Letter 2]

15 Aug 2023

PONE-D-22-24997R2Overemphasis on publications may disadvantage historically excluded groups in STEM before and during COVID-19: a survey-based studyPLOS ONE

Dear Dr. Rowland,

Thank you for submitting your manuscript to PLOS ONE. After careful consideration, we feel that it has merit but does not fully meet PLOS ONE’s publication criteria as it currently stands. Therefore, we invite you to submit a revised version of the manuscript that addresses the points raised during the review process.

 You have done a great job in concluding your project. However, I'd like you to please: Add a section where you address "limitations of the Study" where you can use some of the responses you've written for the previous reviewers. I'd also like you to share your suggestions of improvements if you or anyone else wants to repeat this study or conclude a similar one.You briefly address ESL in your supporting materials. I'd like you to elaborate on ESL and internationals with one or two paragraph in your main text. Over one third of US trainees are internationals. They deal with other marginalizing factors on top of what US-born students do. "Accent Trumps Race" for example. Some come from the "Global South." And, as the last reviewer mentioned, they might hesitate to share their thoughts in a survey because of the cultural differences and hesitations. In the pandemic, and despite the privilege that American students had, many internationals didn't get the chance to see their families, were in exile, and that cannot be ignored. I do not expect you to spend more than a couple of paragraphs on this. I just want you to acknowledge their hardship. You can elaborate on this in the discussion or in the "limitations of the study" for example.Minor revisions:Title: Add “in the United States” at the end of the title. The word “historically” imposes a different meaning to the word “disadvantaged” globally. As a reminder, PLOS ONE is a global platform with readers and authors from across the world.Introduction > Line 3: “hereafter, postdocs” change to “hereafter, trainees”

I understand that the COVID is almost over, and you might think (as you mentioned in an email to me) that the readers might not find this as relevant. I respectfully disagree. You did such a great job in addressing one of the least acknowledged topics. The impact of the COVID is just an example of each intervention for the marginalized/excluded groups you targeted. It will be as relevant as any imposed intervention. To save you time, and as soon as I receive a new revision submission, I will accept it without sending it for review again. In other words, consider this "minor revision" as a "conditionally accepted" letter from me. I look forward to receiving your resubmission. 

We look forward to receiving your revised manuscript.

Kind regards,

Sina Safayi, D.V.M., Ph.D.

Academic Editor

PLOS ONE

Journal Requirements:

Reviewers' comments:

Reviewer's Responses to Questions

**Comments to the Author**

1. If the authors have adequately addressed your comments raised in a previous round of review and you feel that this manuscript is now acceptable for publication, you may indicate that here to bypass the “Comments to the Author” section, enter your conflict of interest statement in the “Confidential to Editor” section, and submit your "Accept" recommendation.

Reviewer #4: (No Response)

2. Is the manuscript technically sound, and do the data support the conclusions?

Reviewer #4: Yes

3. Has the statistical analysis been performed appropriately and rigorously? 

Reviewer #4: I Don't Know

4. Have the authors made all data underlying the findings in their manuscript fully available?

Reviewer #4: Yes

5. Is the manuscript presented in an intelligible fashion and written in standard English?

Reviewer #4: Yes

6. Review Comments to the Author

Reviewer #4: I recommend accepting this manuscript.

The topic and research goals of the “Overemphasis on publications may disadvantage historically excluded groups in STEM before and during COVID-19: a survey-based study” is very relevant. Moreover, The authors have adequately addressed reviewers comments.

I hope their major observation that negative impact of pandemic on writing habits among BIPOC and female graduate students and postdocs inspires discussion and action within higher ed institutions. The compounding effects of social isolation, lack of peer networks and advisor mentorship, institutional writing support, mental health decline and community violence and harm presented during the pandemic can’t be brushed aside as we go into ‘business as usual mode’ in academia. I support the authors call for inclusive metrics for new faculty hiring that doesn’t prioritize publications above all else.

Besides, the central observation, the following results also piqued my interest:

• 40% of graduate students had published before starting graduate school- I wonder if this is standard for STEM disciplines now or an outlier for environmental sciences. This would further create disparity among undergraduates (predominantly historically excluded groups) who don’t have early access to research opportunities.

• While ESL students were the only population to provide text responses on challenges, they weren’t forthcoming on communicating lack of motivation due to pandemic. What percent of these ESL students are international? I wonder if this is a consequence of cultural stigma around reporting mental health decline. I also wonder whether Generative AI tool or writing assistant apps can mitigate writing challenges for ESL groups.

7. PLOS authors have the option to publish the peer review history of their article (what does this mean?). If published, this will include your full peer review and any attached files.

Reviewer #4: No

---

## [Author Response · Author response to Decision Letter 2]

18 Aug 2023

17 August 2023

Dear Dr. Safayi,

We thank you for your quick and thoughtful review of our manuscript (PONE-D-22-24997R2). We have incorporated all of the changes both you and the reviewer suggested and think it clarified and strengthened our manuscript. 

Below we detail point-by-point responses to the comments. 

We look forward to having this paper published in PLOS ONE. Thank you again for the work you put in to helping us get this across the finish line.

Best regards,

Freya Rowland and Kyra Prats, on behalf of all coauthors

—----

PONE-D-22-24997R2

Overemphasis on publications may disadvantage historically excluded groups in STEM before and during COVID-19: a survey-based study

PLOS ONE

Dear Dr. Rowland,

Thank you for submitting your manuscript to PLOS ONE. After careful consideration, we feel that it has merit but does not fully meet PLOS ONE’s publication criteria as it currently stands. Therefore, we invite you to submit a revised version of the manuscript that addresses the points raised during the review process.

You have done a great job in concluding your project. However, I'd like you to please:

Add a section where you address "limitations of the Study" where you can use some of the responses you've written for the previous reviewers. I'd also like you to share your suggestions of improvements if you or anyone else wants to repeat this study or conclude a similar one.

(Answer) We have added a section titled “limitations of the study” to the Discussion to expand upon what we wish we could have measured but didn’t. This expanded our paragraph about intersectionality: 

“Given differences in publishing and research between STEM fields, our results most prominently reflect the experiences of environmental biology trainees; while our findings can inform our understanding of the pandemic’s impact across STEM field, future research is needed to explicitly disentangle the effects of the pandemic on identity in a wide range of STEM fields. We chose an anonymous survey mechanism to maximize and standardize our sample size, but in doing so, we lack the nuance we could have included if our study had also allowed for follow-up interviews. The survey questions were subject to interpretation and reporting differences among respondents. Because of this, our analyses had to be neutral; for example– e.g., we asked trainees whether the pandemic impacted their writing, however we cannot fully conclude it was a negative impact, despite follow-up questions showing a perceived lack of motivation and productivity. Additionally, although ESL trainees did not report effects of the pandemic on writing habits, it could very well be that cultural stigmas around discussing difficulties prevented ESL and international students from answering our questions honestly, especially considering the free responses indicated writing was much more challenging for these trainees (Table 1). The survey also lacked the ability to differentiate between ESL trainees who reside in their home country and international students. Furthermore, caretaker status would have been an informative metric due to school and care center closure during the pandemic, as this likely had a very strong influence on productivity. Future research should disentangle these effects further.”

You briefly address ESL in your supporting materials. I'd like you to elaborate on ESL and internationals with one or two paragraph in your main text. Over one third of US trainees are internationals. They deal with other marginalizing factors on top of what US-born students do. "Accent Trumps Race" for example. Some come from the "Global South." And, as the last reviewer mentioned, they might hesitate to share their thoughts in a survey because of the cultural differences and hesitations. In the pandemic, and despite the privilege that American students had, many internationals didn't get the chance to see their families, were in exile, and that cannot be ignored. I do not expect you to spend more than a couple of paragraphs on this. I just want you to acknowledge their hardship. You can elaborate on this in the discussion or in the "limitations of the study" for example.

(Answer) We agree that the ESL results were compelling, and that is a large part of the reason we included the free responses as a table in the main paper (Table 1). Although the two first-authors are from the U.S. with English as a first language, we’d like to note that two of the co-authors are international trainees with ESL. We have added more to the discussion about the experience of ESL trainees.

We didn’t want to speculate too much about the experiences of trainees, but we agree that highlighting the COVID experience of international students is important. We have added a sentence to the discussion: “In the case of trainees with ESL – many of whom are likely international trainees – the pandemic prevented many from being able to see their families. However, these trainees already were physically distanced from their families even prior to the pandemic, which could explain why they were less likely to report the pandemic affected their writing habits given that they could have already been experiencing isolation.”

We also have a paragraph about ESL trainees in the discussion highlighting how much harder writing is in general:

“Trainees with ESL had slightly fewer publications than trainees with English as a first language. Although these trends were weak, many respondents explicitly mentioned the struggles of English as the default language of science in the open response section (Table 1). Research shows that trainees with ESL face enormous obstacles to scientific publishing [35] but writing intervention programs have helped this group gain confidence and skills in scientific writing [8]. Universities should invest in extra resources for trainees with ESL, including discipline-specific writing courses [36,37].”

Finally, we discuss the ways in which we could have expanded on the survey questions related to the ESL experience in our new “Limitations of the study” section:

“Additionally, although ESL trainees did not report effects of the pandemic on writing habits, it could very well be that cultural stigmas around discussing difficulties prevented ESL trainees from answering our questions honestly, especially considering the free responses indicated writing in general was much more challenging for these trainees (Table 1). The survey also lacked the ability to differentiate between ESL trainees who reside in their home country and international students.”

Minor revisions:

Title: Add “in the United States” at the end of the title. The word “historically” imposes a different meaning to the word “disadvantaged” globally. As a reminder, PLOS ONE is a global platform with readers and authors from across the world.

(Answer) We have edited the title to say “North America” instead of the United States because we sent targeted emails to both U.S. and Canadian institutions. However, it is important to note that we likely had responses from worldwide. Although we did not track locations of responses in our main survey, the option gift card drawing indicated responses from Europe and Australia as well.

Introduction > Line 3: “hereafter, postdocs” change to “hereafter, trainees”

(Answer) Edited.

I understand that the COVID is almost over, and you might think (as you mentioned in an email to me) that the readers might not find this as relevant. I respectfully disagree. You did such a great job in addressing one of the least acknowledged topics. The impact of the COVID is just an example of each intervention for the marginalized/excluded groups you targeted. It will be as relevant as any imposed intervention. To save you time, and as soon as I receive a new revision submission, I will accept it without sending it for review again. In other words, consider this "minor revision" as a "conditionally accepted" letter from me. I look forward to receiving your resubmission. 

(Answer) Thank you for expediting our manuscript. The timeliness is a top concern of the authors and we are relieved to hear you believe it is still relevant.

We look forward to receiving your revised manuscript.

Kind regards,

Sina Safayi, D.V.M., Ph.D.

Academic Editor

PLOS ONE

Journal Requirements:

Reviewers' comments:

Reviewer's Responses to Questions

Comments to the Author

1. If the authors have adequately addressed your comments raised in a previous round of review and you feel that this manuscript is now acceptable for publication, you may indicate that here to bypass the “Comments to the Author” section, enter your conflict of interest statement in the “Confidential to Editor” section, and submit your "Accept" recommendation.

Reviewer #4: (No Response)

2. Is the manuscript technically sound, and do the data support the conclusions?

Reviewer #4: Yes

3. Has the statistical analysis been performed appropriately and rigorously?

Reviewer #4: I Don't Know

4. Have the authors made all data underlying the findings in their manuscript fully available?

Reviewer #4: Yes

5. Is the manuscript presented in an intelligible fashion and written in standard English?

Reviewer #4: Yes

6. Review Comments to the Author

Reviewer #4: I recommend accepting this manuscript.

The topic and research goals of the “Overemphasis on publications may disadvantage historically excluded groups in STEM before and during COVID-19: a survey-based study” is very relevant. Moreover, The authors have adequately addressed reviewers comments.

I hope their major observation that negative impact of pandemic on writing habits among BIPOC and female graduate students and postdocs inspires discussion and action within higher ed institutions. The compounding effects of social isolation, lack of peer networks and advisor mentorship, institutional writing support, mental health decline and community violence and harm presented during the pandemic can’t be brushed aside as we go into ‘business as usual mode’ in academia. I support the authors call for inclusive metrics for new faculty hiring that doesn’t prioritize publications above all else.

(Answer) Thank you! This means a lot that you connected with the message of our manuscript.

Besides, the central observation, the following results also piqued my interest:

• 40% of graduate students had published before starting graduate school- I wonder if this is standard for STEM disciplines now or an outlier for environmental sciences. This would further create disparity among undergraduates (predominantly historically excluded groups) who don’t have early access to research opportunities.

(Answer) That is a good point. Many people from HEGs would not likely have access to research opportunities or the support necessary to publish. A publication is certainly helpful for getting into top graduate programs - it would be really interesting to see how pre-graduate school publications affect admissions at top schools and access to scholarship and fellowship opportunities. Unfortunately, we don’t have any location information for our respondents.

• While ESL students were the only population to provide text responses on challenges, they weren’t forthcoming on communicating lack of motivation due to pandemic. What percent of these ESL students are international? I wonder if this is a consequence of cultural stigma around reporting mental health decline. I also wonder whether Generative AI tool or writing assistant apps can mitigate writing challenges for ESL groups.

(Answer) We agree that this is interesting. The survey was anonymous, but it could very well be that cultural stigmas around reporting difficulties prevented ESL and international students from answering our questions honestly. We have added some of this to the discussion: “We chose an anonymous survey mechanism to maximize and standardize our sample size, but in doing so, we lack the nuance we could have included if our study had also allowed for follow-up interviews. The survey questions were subject to interpretation and reporting differences among respondents. Because of this, our analyses had to be neutral – e.g., we asked trainees whether the pandemic impacted their writing, however we cannot conclude it was a negative impact. Additionally, although ESL trainees did not report effects of the pandemic on writing habits, it could very well be that cultural stigmas around discussing difficulties prevented ESL and international students from answering our questions honestly – especially considering the free responses indicated writing was much more challenging for these trainees (Table 1). The survey also lacked the ability to differentiate between ESL trainees who reside in their home country and international students.”

(Answer) Regarding the proportion of ESL students that are international, that was a discussion we had early on. Being an international student in no way dictates that you are communicating in a second or third language (e.g., a Canadian or UK resident in the U.S.), and ESL students can also be from the same country but have parents with a different primary language. Unfortunately our survey is not able to tease this apart, but there are a large portion of Americans that come from Spanish-speaking households, for example, so we chose to leave it as ESL but added a sentence to the discussion (see above) about how we were unable to separate ESL and international trainees, but it would be very useful for future studies.

Interesting point about AI perhaps being helpful for ESL trainees. We lack any sort of data currently on the directions of AI and how it will be incorporated into academic writing, but hopefully it is useful and makes the process a bit easier on those whose first language is not English.

7. PLOS authors have the option to publish the peer review history of their article (what does this mean?). If published, this will include your full peer review and any attached files.

Do you want your identity to be public for this peer review? For information about this choice, including consent withdrawal, please see our Privacy Policy.

Reviewer #4: No

---

## [Editor Report · Decision Letter 3]

23 Aug 2023

Overemphasis on publications may disadvantage historically excluded groups in STEM before and during COVID-19: a North American survey-based study

PONE-D-22-24997R3

Dear Dr. Rowland,

We’re pleased to inform you that your manuscript has been judged scientifically suitable for publication and will be formally accepted for publication once it meets all outstanding technical requirements.

Kind regards,

Sina Safayi, D.V.M., Ph.D.

Academic Editor

PLOS ONE
---

## [Editor Report · Acceptance letter]

31 Aug 2023

PONE-D-22-24997R3 

Overemphasis on publications may disadvantage historically excluded groups in STEM before and during COVID-19: a North American survey-based study 

Dear Dr. Rowland:

I'm pleased to inform you that your manuscript has been deemed suitable for publication in PLOS ONE. Congratulations! Your manuscript is now with our production department. 

Kind regards, 

on behalf of

Dr. Sina Safayi 

Academic Editor

PLOS ONE